# An Explainable Machine-Learning Model for Compensatory Reserve Measurement: Methods for Feature Selection and the Effects of Subject Variability

**DOI:** 10.3390/bioengineering10050612

**Published:** 2023-05-19

**Authors:** Carlos N. Bedolla, Jose M. Gonzalez, Saul J. Vega, Víctor A. Convertino, Eric J. Snider

**Affiliations:** 1U.S. Army Institute of Surgical Research, JBSA Fort Sam Houston, San Antonio, TX 78234, USA; 2Department of Medicine, Uniformed Services University, Bethesda, MD 20814, USA; 3Department of Emergency Medicine, University of Texas Health, San Antonio, TX 78229, USA; 4Department of Biomedical Engineering, University of Texas Health, San Antonio, TX 78249, USA

**Keywords:** compensatory mechanisms, machine learning, feature extraction, signal processing, personalized medicine, lower body negative pressure

## Abstract

Tracking vital signs accurately is critical for triaging a patient and ensuring timely therapeutic intervention. The patient’s status is often clouded by compensatory mechanisms that can mask injury severity. The compensatory reserve measurement (*CRM*) is a triaging tool derived from an arterial waveform that has been shown to allow for earlier detection of hemorrhagic shock. However, the deep-learning artificial neural networks developed for its estimation do not explain how specific arterial waveform elements lead to predicting *CRM* due to the large number of parameters needed to tune these models. Alternatively, we investigate how classical machine-learning models driven by specific features extracted from the arterial waveform can be used to estimate *CRM*. More than 50 features were extracted from human arterial blood pressure data sets collected during simulated hypovolemic shock resulting from exposure to progressive levels of lower body negative pressure. A bagged decision tree design using the ten most significant features was selected as optimal for *CRM* estimation. This resulted in an average root mean squared error in all test data of 0.171, similar to the error for a deep-learning *CRM* algorithm at 0.159. By separating the dataset into sub-groups based on the severity of simulated hypovolemic shock withstood, large subject variability was observed, and the key features identified for these sub-groups differed. This methodology could allow for the identification of unique features and machine-learning models to differentiate individuals with good compensatory mechanisms against hypovolemia from those that might be poor compensators, leading to improved triage of trauma patients and ultimately enhancing military and emergency medicine.

## 1. Introduction

Monitoring vital signs, often via means of physiological sensors, is critical throughout patient care, but there remain shortcomings with triage decision support. Accurate patient status assessment can only begin after sensor placement, which for invasive line placement can preclude data collection in the pre-hospital setting. This challenge extends particularly to military settings, where remote, austere environments with less skilled medical personnel can further compromise efficacious sensor placement [1,2]. An ongoing research and development goal is to develop wearable healthcare devices capable of improving these shortcomings by allowing watches, adhesive e-tattoos, or ingestible sensors to record data at the point of injury and continue through higher echelons of care [3,4,5]. 

Another shortcoming of current physiological monitors is the inability to provide the caregiver with the most complete picture of a patient’s clinical status which, in turn, can lead to inadequate, inappropriate, or ineffective medical intervention. This problem can extend to wearable healthcare devices as some physiological parameters can be influenced by fear or pain, as well as the high variability of other factors, such as degrees of injury severity and choice of interventions. The complexity of physiology of a trauma patient is further complicated by the high inter- and intra-patient variability with compensatory mechanisms that can mask shock symptoms until life-saving interventions are less effective [6]. Physiological compensation mechanisms protect oxygen delivery to vital organs, primarily by redistributing blood flow to the brain and heart via sympathetic vasoconstriction and increased extraction of oxygen from blood [7]. To address the development of monitoring capabilities for accurate assessment of the complex nature of physiological compensation, recent efforts have focused on the application of artificial intelligence (AI) or machine-learning (ML) algorithms to better interpret sensor data. This has been demonstrated for estimating core body temperature [8], blood pressure [9,10,11,12], and compensatory reserve [6,13,14]. While these methods allow for tracking and estimating metrics and values that are otherwise not possible from raw data streams, oftentimes decisions and methods employed specifically via AI models cannot be easily explained due to the large number of parameters (sometimes millions) that are required by these algorithms [15,16]. This is a critical challenge from a regulatory perspective but also for medical adoption of the technology, as recent studies have shown that explainable AI models can improve trust and reliance on AI by medical personnel [17].

In an effort to avoid the creation of a “black box” AI algorithm, we assessed extracting measurable features of arterial waveforms to calculate a triaging metric known as the Compensatory Reserve Measurement (*CRM*), which aims to represent the biophysics that underly cardiac and peripheral vascular mechanisms of compensation for clinical conditions of central hypovolemia. Signal processing with the extraction of specific waveform characteristics has been used to identify individual parameters, such as traditional systolic and diastolic pressure, as well as millions of complex combinatorial features [18]. The feature extraction approach used in this present study allows for a more explainable algorithm, as the key waveform features that are responsible for tracking *CRM* can be specifically identified. Therefore, through extracting waveform features from arterial waveform data sets, we hypothesized that the compensatory reserve measurement can be tracked via a classical machine-learning model just as well as more complex “black box” AI models. Furthermore, feature extraction methods have the potential to reveal sub-populations as determined by heterogeneity in the identified waveform features needed to predict *CRM*.

### 1.1. Overview of Feature Extraction

Feature extraction is the process of identifying specific repeated characteristics or patterns that are observed in a signal. Features can be absolute magnitudes, relative time, or percentage differences between features, algebraic combinations of multiple features, as well as many other approaches. Extracted features can lessen the data input into an algorithm as a single feature can represent thousands of raw waveform data points of the original signal [19]. These techniques have been used extensively to develop machine-learning algorithms capable of analyzing a variety of biomedical signals. For example, a previous study used wavelet-based feature extraction methods in developing algorithms that can differentiate between a subject’s “resting” and “thinking” status from electroencephalography signals with 98% accuracy [20]. In another study, researchers extracted 3022 individual features from arterial waveform data and employed combinatory techniques to produce more than 2 million features to successfully develop an ML model for detecting hypotension during intensive care and surgical procedures [18]. Finally, a previous study developed linear regression models using nine features extracted from arterial waveform data to estimate *CRM*. In that study, all of these features were not required, as ultimately, a single feature was able to linearly correlate to *CRM* [21]. However, this study used a limited dataset of 13 human subjects exposed to low levels of lower body negative pressure, resulting in less subject variability in the data set. 

### 1.2. Compensatory Reserve Measurement

The Compensatory Reserve Measurement is a triaging metric previously developed at the US Army Institute of Surgical Research that estimates the level of physiological decompensation in an individual experiencing hypovolemia [22]. This compensatory index provides a value on a scale of 0 to 1 (or 0% to 100%) that reflects the overall physiological compensation status, where 0 represents the threshold of decompensated shock. The idea behind this measurement is to provide a capability for a clinical caregiver to predict with sufficient time the imminent onset of overt shock in a patient who might not otherwise show signs of hemodynamic decompensation based on standard vital signs. The *CRM* was previously defined by Moulton et al. as follows [13]:(1)$$CRM=1-\frac{BLV}{BLV_{HDD}},$$
where *BLV* is the volume of blood a patient has lost, and *BLV_HDD_* is the volume of blood loss required for the same patient to reach hemodynamic decompensation (*HDD*), i.e., to exhaust the body’s compensatory reserve. Since neither of these volumes can be known with certainty in real-world scenarios, mathematical algorithms have been developed instead to estimate a *CRM* value from features found in the patient’s arterial waveform [13,22]. 

The development of these algorithms requires physiological data obtained during hemorrhage to the point of *HDD*. Since experimenting on a human subject by hemorrhaging them to *HDD* would be dangerous and unethical, a more practical method has been used since the 1960s to safely simulate the experience of hemorrhaging large volumes of blood. In this model, the lower section of the body of a healthy research subject is placed inside a negative pressure chamber sealed around the waist. By applying increasing levels of lower-body negative pressure (*LBNP*) to the subject, their blood volume is progressively redistributed to the lower portions of the body, which results in central hypovolemia, thus simulating the physiological effects of hemorrhage. Using this technique, the subject can be brought to the point of hemodynamic instability safely and in a controlled fashion [23]. 

Using *LBNP* as a model for simulated blood loss, the *CRM* can then be estimated by the following formula [13]:(2)$$CRM=1-\frac{LBNP}{LBNP_{HDD}},$$
where *LBNP* is the level of negative pressure a subject is experiencing at any given time, and *LBNP_HDD_* is the level of negative pressure at which the same subject reaches *HDD*.

Algorithms developed to estimate a *CRM* value from *LBNP* data attempt to match features found on a relevant physiological waveform, such as pulse oximetry and/or arterial pressure, to a subject’s level of *HDD* [13,14]. In this technique, a reference *CRM* of 1.0 is assigned to the baseline *LBNP* level (i.e., zero negative pressure), and a *CRM* of 0.0 is assigned to the *LBNP* level at which a subject experiences *HDD*. A machine-learning model is then trained to match patterns and features found in the physiological waveform to each *LBNP* level found between those two reference points. The goal is for the resulting algorithm to be able to predict the instantaneous relative *LBNP* level of an individual and thus estimate their level of *HDD* on a scale from 0.0 to 1.0 based on a sample of a relevant physiological waveform, without any additional knowledge of that individual’s physiological status.

## 2. Materials and Methods

### 2.1. Retrospective Analysis of Lower Body Negative Pressure Datasets

*LBNP* datasets were previously generated during human research studies performed at the US Army Institute of Surgical Research [13]. These prior studies were approved by the required Institutional Review Board committees and followed the guidelines of the Declaration of Helsinki. 

Briefly, during an *LBNP* experimental session, the lower portion of the body of a subject laying in the supine position was placed inside a sealed vacuum chamber. During the protocol, negative pressure was safely and progressively applied to the subject’s lower body from baseline pressure of 0 mmHg down to a minimum of −100 mmHg, in 9 incremental steps that lasted 5 min each (Figure 1A). Each experiment ended when the pressure reached the aforementioned minimum limit, or when the subject reached their own individual *LBNP_HDD_*, at which point the chamber’s pressure was returned to baseline. 

Throughout the *LBNP* experiments photoplethysmography (PPG) waveform data sampled at 500 Hz were recorded using a Finometer PRO blood pressure monitor (Finapres Medical Systems, Amsterdam, The Netherlands) from 218 subjects. The data were further processed and analyzed to develop new machine-learning models to estimate *CRM*; these models will be referred to as CRM-ML throughout.

### 2.2. Pre-Processing Datasets

De-identified data from the *LBNP* subjects were used for analysis in this study and were the foundation for creating an ML algorithm for the prediction of *CRM* at different hypovolemic states. Subjects reached varying final pressure levels of *LBNP* exposure, so each dataset was classified by its maximum *LBNP* step reached prior to *HDD* to determine the sample distribution (Figure 1B). Due to these varying levels of *LBNP* steps reached, data were equally sampled from each *LBNP_HDD_* step to equally account for differences between each *LBNP_HDD_* sub-group in the *CRM*-ML algorithm. However, due to limited subject data in the distribution extremes, datasets for *LBNP_HDD_* at steps 2, 3, and 9 were removed from consideration. Equal number of randomly sampled 4, 5, 6, 7, and 8 *LBNP_HDD_* step subjects were then used. While a total of 218 subjects were available in the dataset, step group 4 had only 12 subjects available, limiting the number of subjects for all other steps to maintain an 8:4 training-to-blind testing ratio. Of these selected subjects, 62% were male and 38% were female, with heights ranging between 152 and 193 cm, weights ranging between 44 and 117 kg, and ages ranging between 18 and 54 years of age. A total of 8 subjects were selected from each *LBNP_HDD_* step group for the training of the CRM-ML algorithm (40 total subjects), and 4 subjects for each *LBNP_HDD_* were held out for blind testing (20 total subjects). 

### 2.3. Feature Extraction Methodology

Features were extracted from the arterial pressure vs. time *LBNP* datasets using MATLAB (v2022b) MathWorks, Natick, MA, USA). First, the wavelength was filtered using a finite impulse response (FIR) window lowpass filter [24]. The filtered data were then analyzed to calculate the first and second derivatives, which were then used to identify the peaks and troughs of the waveform that could assist in locating important landmarks in the signal (Figure 2). For simplicity, the start of the systolic phase of the waveform, or pulse foot, was identified as point “A” and the systolic peak as point “C”. A halfway point between “A” and “C” was identified as “B” to represent the “half-rise” [25]. Lastly, the inflection point after the systolic peak was labeled as point “D”. Inflection points were not always identified in some segments of the arterial waveforms. For consistency, if inflection points were not identified, these arterial waveform segments were excluded from the analysis. In other waveform segments, two or more inflection points were identified after the systolic peak. In these situations, only the first inflection point was tracked for consistency.

These points were then used to calculate additional waveform features selected from previous research efforts by Gupta et al. [21] and Hatib et al. [18]. A total of 54 features were extracted from each waveform as summarized in Table 1. More detailed definitions for these features, including their equations are detailed in Appendix A. These features were collected from each subject dataset for training an ML model and for blind test data to later assess model performance. 

### 2.4. Machine-Learning Models

The Regression Learner Toolbox in MATLAB was used for evaluating a wide range of machine-learning models for estimating *CRM*. Training was performed on 40 total *LBNP* subjects (n = 8 for each *LBNP_HDD_* step). For feature selection, features were ranked using the minimal-redundancy–maximal-relevance (MRMR) criterion, which balances selection of a feature between maximum correlation to the signal and least correlation to other features [26]. To evaluate different ML models, Linear Regression, Fine Tree, Medium Tree, Coarse Tree, Ensemble Bagged Trees, and Ensemble Boosted Trees models were constructed using MATLAB’s Regression Learner Toolbox. The two ensemble models have an eight-leaf size with 30 learners. To find the optimal feature selection, all models were trained with the top 15, 10, 5, and 1 feature(s). Blind testing was conducted using 4 subjects, separate from the training subjects, from each *LBNP* step group (20 subjects total). To identify differences among subject subpopulations, a second approach to model training was used, in which only data from each single *LBNP* step group were used to train a single CRM-ML model, i.e., one model per step group. For these individual models, the same methods for training the general model were used, except only the 8 subjects for that specific *LBNP_HDD_* step group were used during training. Testing was still conducted using all the datasets. 

In order to compare and down-select the optimal machine-learning model and number of features, the training and testing root-mean-squared error (RMSE) and coefficient of determination (R^2^) values were computed. It is important to note that the R^2^ and RMSE values were collected from the Regression Learner Toolbox. The methodology for calculating R^2^ and RMSE values in this toolbox was conducted via fitting *CRM* predictions vs. *CRM* calculated (Equation (2)) values to a perfect regression (y=x), as opposed to the more conventional linear regression (y=mx+b). Thus, throughout, these values obtained from the toolbox will be referred to as perfect R^2^ (or P-R^2^) and perfect RMSE (or P-RMSE) to differentiate them from the traditional regression (R^2^) and RMSE values for the two-parameter regression fit. 

### 2.5. Deep-Learning Model

To compare the performance of the new machine-learning models developed in this study to more complex algorithms, we estimated *CRM* values on the same *LBNP* datasets using a deep-learning model (CRM-DL) with a one-dimensional Convolutional Neural Network [14]. For its development, arterial waveform data sampled at 100 Hz was normalized to a 0–1 range and then segmented into 20 s long samples for model regression learning. The model then learned to perform a linear regression on each data segment to estimate a corresponding *CRM*.

To evaluate the CRM-DL model, we again used the same *LBNP* subject datasets used for testing the CRM-ML models (n = 4 per each *LBNP_HDD_* step group). The result of this process was a series of *CRM* values sampled at 10 s intervals that spanned the full length of the *LBNP* sessions for each experimental subject. Both conventional and perfect linear regression methods were used to calculate RMSE and R^2^ to evaluate the performance of CRM-DL in predicting *CRM* against the calculated *CRM* value (Equation (2)).

The likelihood that during our evaluation of its performance the CRM-DL may have processed instances already learned is noteworthy given that the model was originally trained on the same *LBNP* datasets. This would give the CRM-DL model an unfair advantage when comparing its performance against that of our new classical ML models. However, the exact number of records that overlap the DL training set and our evaluation, if any, is unknown.

## 3. Results

### 3.1. Machine-Learning Model and Number of Features Selection

Different machine-learning model types were trained to predict *CRM* using different numbers of extracted arterial waveform features (Table 2). The ensemble bagged trees model performed the best, with lower P-RMSE and highest P-R^2^ in training, regardless of the number of features used, while linear regression was consistently one of the lowest-performing models for both P-RMSE and P-R^2^. During blind testing, the simpler decision trees performed worse than other ML models, likely indicating an overfitting of training data. Training performance was worsened when only a single feature was considered by the model, but blind testing performance remained similar for most CRM-ML models. During blind testing, the ensemble bagged tree and boosted tree ML model performed well. In general, P-RMSE and P-R^2^ results showed that a higher number of features in ML models were associated with better model performance, especially for training performance. However, the improvement in performance between 15 and 10 features was relatively small, so it was decided to continue testing with the ensemble bagged trees ML model using 10 features.

### 3.2. Effect of LBNP Final Step Reached on Model Performance

Using the 10-feature ensemble bagged-tree ML model, we examined the effect of subjects’ *LBNP* performances (i.e., tolerance) on model accuracy. Representative plots of theoretical *CRM* (Equation (2)) compared to the predicted CRM-ML are displayed in Figure 3 for the different *LBNP_HDD_* step groups. Both perfect and conventional linear regression are highlighted for each. Predictions of *CRM* at specific central blood volume levels (i.e., *LBNP* levels) tracked the general theoretical *CRM* trend but demonstrated a bias toward certain steps in the *LBNP* process and subgroups. In general, CRM-ML predictions were more accurate for conventional regression approaches compared to perfect regression. 

To highlight the difference in model performance for the different subgroups, R^2^ and RMSE were calculated for both regression methods for the blind testing at each step for the all-step training model (n = 4 subjects each). The average results are summarized in Table 3. Overall, the testing on the 6-step subgroup resulted in higher coefficients of determination and lower root mean square errors, using both perfect and standard linear regression methodologies, while the tests on the 4-step subgroup resulted in worse scores on both accounts. 

### 3.3. Differences in Features for Separate Models

To further highlight differences in subject subgroups based on pressure step at *LBNP_HDD_*, five additional CRM-ML models were developed, each one trained on a single sub-group, i.e., one model for each of the 4, 5, 6, 7, and 8 *LBNP_HDD_* subjects. This training approach resulted in ML models with different extracted features being selected as “top 10”. This is detailed in Table 4, where the top ten features, ranked using MRMR, are shown for the different CRM-ML models. The movement of key features in ranking for each single-step trained ML model is evident compared to the all-step CRM-ML model. Certain features, such as HRIP, remained among the top 5 for all individual models, while others traveled significantly, such as sys_rise_area_norm, compared to the model trained with all steps. We postulate that features that persistently ranked among the top ten used in training the models hold significant weight in the prediction of a *CRM*, while some features that only appear at certain step models could be used as a differentiator or predictor of the level of *LBNP* exposure at which an individual may decompensate.

Each single-step trained ML model was blind tested against subjects from within its own *LBNP* group as well as from other step subgroups. For example, the 4-step training model (i.e., the ML model trained only on 4-step subjects) was blind tested on the 4-, 5-, 6-, 7-, and 8-step test data. This was repeated for all models. P-RMSE and P-R^2^ results were plotted in a heat map for comparison purposes, as shown in Figure 4.

The training model with the highest average P-R^2^ (0.81) and the lowest average P-RMSE value (0.13) was the model trained on the 5-step *LBNP_HDD_* datasets. The blind test data that had the highest average P-R^2^ (value range = 0.83–0.92) and the lowest average P-RMSE (value range = 0.09 to 0.13) was the test data at step 6. In general, all the individualized models performed better against their specific datasets at that step.

### 3.4. Deep-Learning Results

CRM-ML model results were compared against the CRM-DL model; the all-step trained CRM-ML model was used for comparison as opposed to the subject-specific models. Calculated “true” *CRM* values and the predictions via both the CRM-DL and the CRM-ML models were plotted against each other to visually assess the differences in their performances. In general, the CRM-ML model had higher accuracy for higher *CRM* values than the CRM-DL model, while both reach similar values of *CRM* in the lower bound. The ML model also tended to overshoot the theoretical *CRM*, while the DL model tended to undershoot it. The plots in Figure 5 show representative blind testing at different steps of both the DL and ML models.

Both the perfect regression and conventional linear regression were compared using both R^2^ and RMSE metrics, as shown in Figure 6. Overall, the CRM-ML model had an average P-R^2^ value higher than CRM-DL (CRM-ML = 0.70 vs. CRM-DL = 0.50), but relatively same P-RMSE values (CRM-ML = 0.171 vs. CRM-DL = 0.165). With conventional linear regression, both the CRM-ML model and CRM-DL had similar average R^2^ (CRM-ML = 0.85 vs. CRM-DL = 0.88) and RMSE values (CRM-ML = 0.168 vs. CRM-DL = 0.165). Performance differences for the different *LBNP_HDD_* steps varied with the 4-step data having the worse P-R^2^ metrics and the 6-step having the strongest.

## 4. Discussion

Advanced monitoring is critical for conducting accurate triage of a patient presenting with significant hemorrhage, especially in scenarios where invasive sensor placement may not be possible, such as in pre-hospital and military settings. To simplify regulatory hurdles and improve medical adoption, algorithms for this purpose should be as simple as possible to meet the needs of emergency medical personnel. In this study, we focused on compensatory reserve measurement, a metric derived from arterial waveform data for accurately tracking physiological decompensation and early detection of hemorrhagic shock. To develop a simpler model as an alternative to the current deep-learning models employed in *CRM* estimation, regression-based machine-learning models were developed using extracted features from the arterial waveform. Simpler ML methods such as this are more explainable, as the individual waveform features critical to calculating *CRM* are identified and may further understand the underlying physiological compensatory mechanisms.

Overall, the feature extraction methodology paired with classical ML models allowed for accurately tracking *CRM*. This approach required extracting ten features from each arterial waveform segment which was performed as a pre-processing setup prior to ML model implementation. However, this feature extraction signal processing could be easily performed during data collection with minimal computing requirements to enable real-time reporting of *CRM* values. Compared to deep-learning models that require processing millions of parameters, the CRM-ML models developed here should perform more computationally efficiently, which may be critical in a resource-limited environment. Feature extraction methodologies have been recently used by Gupta et al. to develop ML models that estimate *CRM* [21]. They used a single parameter model, time from half-rise to dicrotic notch (HRDN), which resulted in R^2^ values of approximately 0.67, lower than our average R^2^ value of 0.85. Different *LBNP* data sets were used by Gupta et al., with different simulated hemorrhage magnitudes and a lower subject number of only n = 13 subjects. The differences in data sets, reduced number of subjects, and the increase from a single extracted feature to 10 features in CRM-ML likely account for our increased R^2^ performance compared to Gupta. It is worth noting that the time from half-rise to inflection point (HRIP) feature used in this study was the top correlative feature in the CRM-ML model and is analogous to the top-performing feature (HRDN) identified by Gupta.

It is important to note that bagged decision tree models have advantages and disadvantages compared to other ML/DL approaches. The crucial advantage of bagged decision tree models is that multiple models are bootstrapped so that not one model drives predictions, allowing for a more robust ensemble. This removes bias since there are multiple models that can agree or disagree creating a more generalized predictive algorithm [27,28]. However, bagging multiple models into a more robust ensemble can lead to losing some interpretability compared to a single model. This is exacerbated in training situations where there are rare, underrepresented events that may be lost in an ensemble, generalized model approach. However, we think this is less of a concern for a use case, such as measuring compensatory status, which needs to be applicable to a large subject population. Another possible disadvantage, depending on the size of the training data, is that training ensemble bagged trees may require a high computational burden [27,28,29]. However, the computational time required to calculate a single prediction is much lower than that consumed for model training and is still much less intense compared to most deep-learning models.

Next, CRM-ML performance was contrasted against the CRM-DL model for predicting *CRM* [14]. Perfect and conventional regression RMSE results were very comparable between the CRM-ML and CRM-DL models, highlighting that the simpler approach used here may be sufficient for tracking *CRM*. However, larger differences were more evident between the two models when evaluating P-R^2^ values (0.70 for CRM-ML vs. 0.50 for CRM-DL). That said, the CRM-DL fit the data worse when subjects did not reach lower pressure steps (*LBNP_HDD_* step 4 P-R^2^ = 0.37 vs. step 8 P-R^2^ = 0.56), just like CRM-ML struggled in these same situations (*LBNP_HDD_* step 4 P-R^2^ = 0.52 vs. step 8 P-R^2^ = 0.74). Of note, when the goodness of fit was evaluated via standard linear regression, the CRM-DL model performed more comparable to CRM-ML (DL R^2^ = 0.88 vs. ML R^2^ = 0.85). This suggests that the deep-learning model can be adjusted to better match calculated *CRM*, but it is not predicting at a 1:1 correlation without a slope, intercept adjustment. One major limitation of this part of this study, however, is that the CRM-DL model was trained previously [14] using all of the *LBNP* step data, meaning the test data used in this study was not completely blind to the CRM-DL algorithm.

An unexpected trend was detected based on the maximum hemodynamic decompensation threshold reached by the subject, presented as the final *LBNP* pressure step reached. The generalized CRM-ML models struggled to perform accurately across each *LBNP* step sub-population, so models were trained for each individual *LBNP* step sub-population. In doing so, the top 10 features selected for each CRM-ML model differed greatly across the individual *LBNP* step models. While features such as HRIP were prevalent in the top features for all CRM-ML *LBNP_HDD_* models, others were specific to certain models, as highlighted in Table 3. Subpopulations have been previously reported from these *LBNP* datasets, referred to as high- and low-tolerant subjects based on exceeding a certain *LBNP* exposure level [7,30]. We chose to not focus on these two sub-groups but instead break the data into more granular subsets to highlight potential differences across subgroups. This may suggest there are more subgroups in these datasets. Furthermore, the variations in features extracted via the individualized models may identify physiologically pertinent differences that can be used to understand variability in the patient’s compensatory reserve capacity. For instance, if these different CRM-ML models are utilized to predict *CRM* in real time, tracking which model best fits the patient may allow for the prediction of the compensatory capacity before hemodynamic decompensation is reached. More analysis and follow-up studies are needed for further teasing out these differences, including a more thorough understanding of the underlying physiology behind the extracted features.

The next steps for this work will be three-fold. First, the existing CRM-ML models will be used with additional data sets for measuring *CRM* compared to the CRM-DL model to further assess model performance. This includes datasets for patients undergoing surgery and animal data with deeper levels of hemorrhagic shock to evaluate performance outside the scope of current CRM-DL training methodologies. Second, the CRM-ML model will be further improved by evaluating additional extracted features, combinations of features, and trends in features as inputs to the model. This may reveal additional, arterial waveform characteristics distinctive of each subject population, and subgroups, and/or further improve model accuracy. Lastly, this feature extraction methodology will be applied to other medical applications, including tracking *CRM* with novel wearable sensors, and identifying other metrics for tracking shock after decompensation and as inputs for guiding hemorrhage fluid resuscitation.

## 5. Conclusions

In conclusion, algorithms for advanced monitoring have the potential to improve medical care at the point of injury by enabling medical providers to have more information to diagnose and provide treatment when it is more effective. Here, we focused on feature extraction methodologies for tracking *CRM* using 10 waveform features as opposed to millions of tuned AI model parameters. Overall, model performance was similar to a more complex deep-learning model, and by tracking extracted features, differences were identified in subgroups in the data sets. Overall, this machine-learning approach can create a more explainable model, a critical step towards the adoption of advanced monitoring technologies in emergency and military medicine.

## Figures and Tables

**Figure 1 bioengineering-10-00612-f001:**
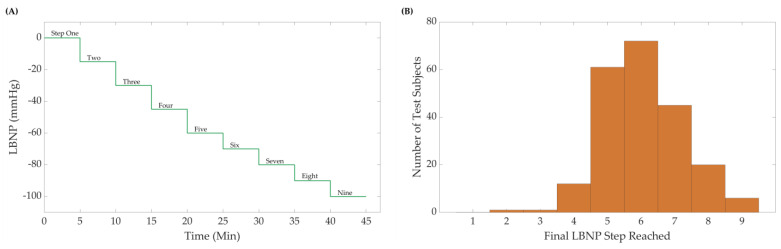
(**A**) The Lower Body Negative Pressure (*LBNP*) step profile and subject distribution. A maximum of nine negative steps were used, including the initial baseline starting pressure. Subjects were removed from *LBNP* at hemodynamic decompensation or after the final pressure step (**B**) Sample distribution (N = 218 subjects) for the *LBNP* pressure where hemodynamic decompensation was reached.

**Figure 2 bioengineering-10-00612-f002:**
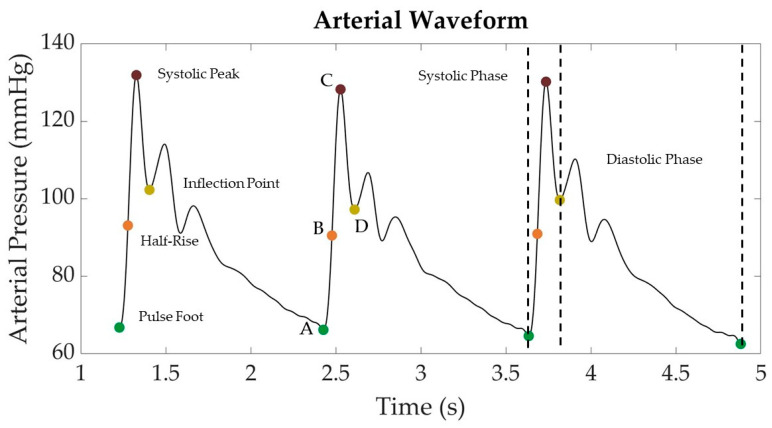
Representative arterial waveform segment from *LBNP* dataset with key landmarks identified. Landmarks include the pulse foot (“A”), systolic peak (“C”), half-rise between pulse foot and systolic peak (“B”), and the post-systolic inflection point (“D”).

**Figure 3 bioengineering-10-00612-f003:**
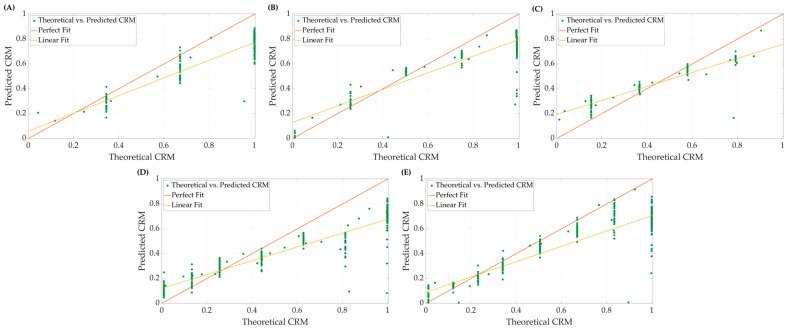
Representative plots comparing perfect fit and linear fit regression models for theoretical vs. predicted *CRM* at (**A**) 4, (**B**) 5, (**C**) 6, (**D**) 7, and (**E**) 8 final *LBNP* step subjects.

**Figure 4 bioengineering-10-00612-f004:**
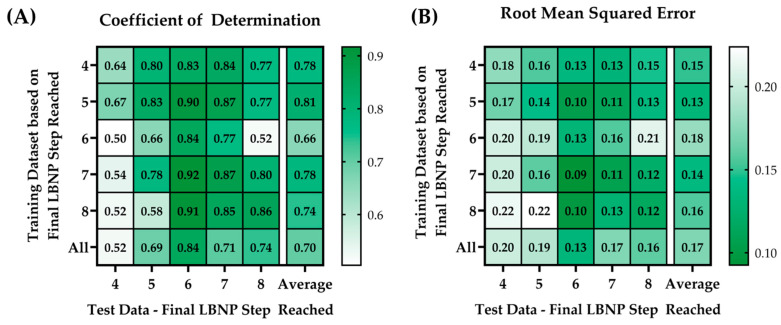
Heat map matrices of the different trained ML models (rows) vs. the blind test data (columns) showing results for (**A**) perfect coefficient of determinations (P-R^2^) and (**B**) perfect root mean squared errors (P-RMSE).

**Figure 5 bioengineering-10-00612-f005:**
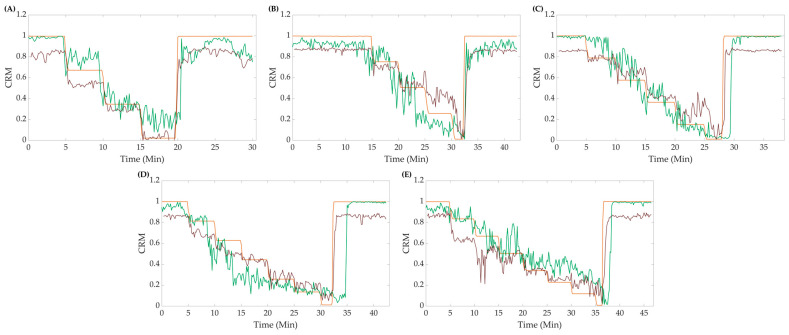
Representative data for (**A**) 4, (**B**) 5, (**C**) 6, (**D**) 7, and (**E**) 8 final *LBNP* step subjects comparing the CRM-ML model (green line), CRM-DL model (red line), and calculated *CRM* (orange line).

**Figure 6 bioengineering-10-00612-f006:**
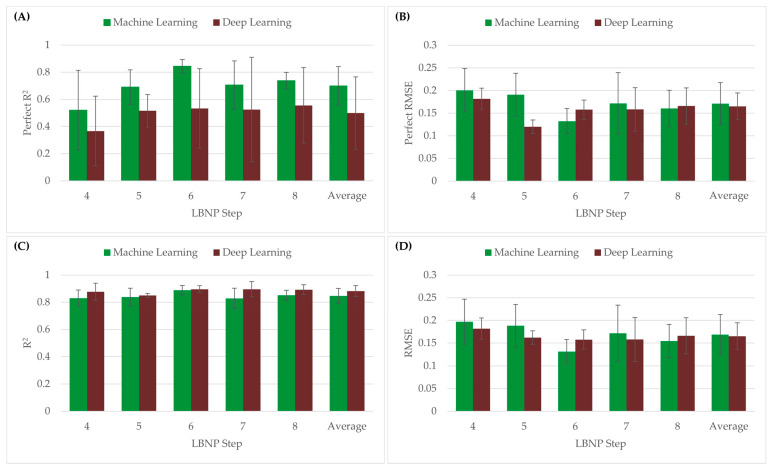
(**A**) P-R^2^ values and (**B**) P-RMSE of CRM-ML and CRM-DL models using the perfect regression method. (**C**) R^2^ and (**D**) RMSE values of CRM-ML and CRM-DL models using the conventional linear regression method.

**Table 1 bioengineering-10-00612-t001:** Summary of the 54 distinct features extracted from each *LBNP* waveform. Appendix A contains additional descriptions for each individual feature. Features calculated by subtracting the pressure value at the inflection point are referred to as “NODIA”.

Feature Types	Description	Number of Features
Individual Features	Features from the arterial waveform consist of standard waveform measurements (such as pulse pressure [PP] and peak-to-peak interval [PPI]).	7
Time Duration Features	Duration of certain phases of the arterial waveform (such as time from half-rise to inflection point [HRIP] and duration of the systolic phase [t_sys]).	6
Average Pressure Features	Average pressures of different arterial waveform phases.	5
Area Under the Curve Features	Area under the curve of different waveform phases.	5
Normalized Features	Area features normalized by the number of samples in the waveform and phases.	10
NODIA Features	Area under the curve and normalized features subtracted by the waveform value of the inflection point.	18
Slope Features	Average slope of different phases of the arterial waveform.	3

**Table 2 bioengineering-10-00612-t002:** Training and testing results for different classical machine-learning models trained using the top 1, 5, 10, or 15 extracted features. Performance results are shown for perfect RMSE and R^2^ values for each model. Green heat map overlay is set between the minimum and maximum value to indicate better performance.

15 Features				
	Training		Testing	
Model	P-RMSE	P-R^2^	P-RMSE	P-R^2^
Linear Regression	0.17865	0.74	0.17766	0.73
Fine Tree	0.10133	0.92	0.23274	0.54
Medium Tree	0.10176	0.92	0.21663	0.6
Coarse Tree	0.10957	0.9	0.20097	0.66
Boosted Tree	0.14322	0.83	0.15966	0.78
Bagged Tree	0.07762	0.95	0.17133	0.75
10 Features				
	Training		Testing	
Model	P-RMSE	P-R^2^	P-RMSE	P-R^2^
Linear Regression	0.17865	0.74	0.18431	0.71
Fine Tree	0.10133	0.92	0.22312	0.58
Medium Tree	0.10306	0.91	0.20833	0.63
Coarse Tree	0.11149	0.9	0.19978	0.66
Boosted Tree	0.14582	0.83	0.16202	0.78
Bagged Tree	0.079094	0.95	0.17442	0.74
5 Features				
	Training		Testing	
Model	P-RMSE	P-R^2^	P-RMSE	P-R^2^
Linear Regression	0.1948	0.69	0.17942	0.73
Fine Tree	0.12501	0.87	0.2225	0.58
Medium Tree	0.12229	0.88	0.20916	0.63
Coarse Tree	0.12754	0.87	0.1925	0.68
Boosted Tree	0.15704	0.8	0.16573	0.77
Bagged Tree	0.1035	0.91	0.17708	0.73
1 Feature				
	Training		Testing	
Model	P-RMSE	P-R^2^	P-RMSE	P-R^2^
Linear Regression	0.19957	0.68	0.18441	0.71
Fine Tree	0.20607	0.66	0.19594	0.67
Medium Tree	0.18954	0.71	0.17926	0.73
Coarse Tree	0.18142	0.73	0.16995	0.75
Boosted Tree	0.17921	0.74	0.17351	0.74
Bagged Tree	0.18796	0.71	0.1774	0.73

**Table 3 bioengineering-10-00612-t003:** Coefficients of determination and root mean squared error performance metrics using the CRM-ML model to predict *CRM* for each *LBNP_HDD_* sub-group. Results are shown for perfect regression (P-R^2^ and P-RMSE) and traditional linear regression (R^2^ and RMSE).

Step Subgroup	P-R^2^	P-RMSE	R^2^	RMSE
4	0.52	0.20	0.83	0.20
5	0.69	0.19	0.84	0.19
6	0.84	0.13	0.89	0.13
7	0.71	0.17	0.83	0.17
8	0.74	0.16	0.85	0.15

**Table 4 bioengineering-10-00612-t004:** Summary of ten highest ranked features for an ML model trained with all data or exclusively the 8, 7, 6, 5, or 4 final *LBNP* step subgroup. Superscript values for each feature indicate how much higher (positive/green), lower (negative/red), or no change (equal sign) the specific extracted feature’s importance in ranking shifted relative to the all-steps CRM-ML model. More information on how each of the features was calculated is described in Appendix A.

	All Steps	8 Steps	7 Steps	6 Steps	5 Steps	4 Steps
Rank	Feature	Feature	Feature	Feature	Feature	Feature
1	HRIP	sys_rise_area_norm^+52^	sys_rise_area_norm^+52^	HRIP^=^	sys_rise_area_norm^+52^	HRIP^=^
2	dec_area_nor	SI^+16^	sys_rise_area_nodia^+18^	avg_sys_rise^+35^	slope_desc_sys^+5^	avg_dia^+33^
3	t_sys_rise	t_sys_rise^=^	t_sys_rise^=^	avg_sys_rise_nodia^+10^	avg_sys_nodia^+11^	avg_sys_dec_nodia^+1^
4	avg_sys_dec_nodia	avg_sys_dec_nodia^=^	HRIP^−3^	PP^+6^	dec_area_nodia^+2^	sys_rise_area_norm^+49^
5	avg_dia_nodia	HRIP^−4^	PP^+5^	t_sys_rise^−2^	HRIP^−4^	slope_sys^+12^
6	dec_area_nodia	dia_area_nodia^+9^	dec_area_nodia^=^	sys_area_norm^+16^	avg_sys_dec_nodia^−2^	t_sys_rise^−3^
7	slope_desc_sys	slope_dia^+1^	sys_area^+2^	sys_dec_area^+16^	t_sys_rise^−4^	t_sys^+23^
8	slope_dia	slope_desc_sys^−1^	avg_sys_dec_nodia^−4^	slope_dia^=^	slope_dia^=^	slope_desc_sys^−1^
9	sys_area	sys_area_nodia^+11^	slope_sys^+8^	pp_area_nodia^+10^	PP^+1^	sys_area^=^
10	PP	pp_area_nodia^+9^	slope_desc_sys^−3^	avg_sys_nodia^+4^	PPI^+1^	dec_area_nodia^−4^

## Data Availability

The data presented in this study are not publicly available because they have been collected and maintained in a government-controlled database that is located at the US Army Institute of Surgical Research. As such, these data can be made available through the development of a Cooperative Research & Development Agreement (CRADA) with the corresponding author.

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
