# Peer review of "An Explainable Machine-Learning Model for Compensatory Reserve Measurement: Methods for Feature Selection and the Effects of Subject Variability"

_bioengineering, 2023, doi:10.3390/bioengineering10050612_

Round 1

Reviewer 1 Report

In this work, An explainable machine learning model for compensatory reserve measurement was proposed. The mian work includes that the classical machine-learning models driven by specific features extracted from the arterial waveform was used to estimate compensatory reserve measurement. The results highlight that the proposed methodology could allow for identification of unique features and machine-learning models to differentiate individuals with good compensatory mechanisms against hypovolemia from those that might be poor compensators, leading to improved triage of trauma patients, ultimately enhancing military and emergency medicine. Alghotugh I have a  concern, that is the repeatability, accuracy, reliability, and applicability of the research results in this article, the research is interesting and worth publishing. 

Author Response

  1. In this work, An explainable machine learning model for compensatory reserve measurement was proposed. The mian work includes that the classical machine-learning models driven by specific features extracted from the arterial waveform was used to estimate compensatory reserve measurement. The results highlight that the proposed methodology could allow for identification of unique features and machine-learning models to differentiate individuals with good compensatory mechanisms against hypovolemia from those that might be poor compensators, leading to improved triage of trauma patients, ultimately enhancing military and emergency medicine. Alghotugh I have a concern, that is the repeatability, accuracy, reliability, and applicability of the research results in this article, the research is interesting and worth publishing.

Thanks for reviewing our manuscript for publication and we appreciate the summary about our work. If there are any edits we can make with regard to the applicability, reliability and other concerns, we would be happy to address these with the reviewer.

Reviewer 2 Report

88 missing bracker in 19 

Authors present an interesting article using Compensatory Reserve Measurement (CRM) using machine learning model. There are several challenges and opportunities in this methodology that if authors clarify, the reader could benefit more of the presented investigation, for example:

Can the authors comment on the issue of Limited and heterogeneous data? Collecting CRM data can be challenging due to the need for invasive or specialized measurements. Additionally, the data may vary across individuals, making it difficult to create a comprehensive and representative dataset.

How does the authors assess a resonable interpretability? Machine learning models used in CRM may produce accurate predictions, but their black-box nature can make it challenging to interpret the underlying factors contributing to those predictions. This can be a concern, especially in healthcare settings where interpretability and explainability are crucial.

Can the authors comment on new feature selection or the given selection is absolute? Determining the most relevant features or physiological parameters to include in the model can be complex. It requires a deep understanding of the underlying physiology and may involve domain experts to guide the feature selection process. This is important when looking at the differences

 0.171, vs 0.159, close enough, but still room for improvement.

Can the authors give comments on generalization? Ensuring that a machine learning model trained on a specific CRM dataset can generalize well to different populations, demographics, or clinical scenarios is crucial. The model needs to account for the variability across individuals and adapt to new data distributions.

In terms of the methodology, bagged decision trees have several advantages, such as reducing overfitting and improving prediction accuracy, however there are also some disadvantages to consider, which would be interesting to hear the thoughts of the authors for the present investigation and scope.

How the Increased complexity can be addressed? Bagging can increase the complexity of the model, as it involves training multiple decision trees and combining their predictions. This can make the model more difficult to interpret and understand, which can be a disadvantage in some applications.

Can the authors comment on the computationally intensive nature of the method particularly for a militar setting? Bagging can be computationally intensive, especially when dealing with large datasets or a large number of decision trees. This can make the training and prediction time longer, which can be a disadvantage in real-time applications.

What are the solutions for bias reduction limitations? While bagging can reduce variance and improve prediction accuracy, it has limited impact on reducing bias. This means that if the underlying decision tree model is biased, the bagged model will also be biased.

For the blood pressure data, how can we assess the balance of the data? Bagging can be less effective for imbalanced datasets, where one class is underrepresented. This is because the decision trees may focus more on the majority class, leading to poor predictions for the minority class.

Can the authors give some guidelines on Limited interpretability? The combination of multiple decision trees in a bagged model can make it difficult to interpret the importance of individual features and their contribution to the final prediction. This can be a disadvantage in applications where interpretability is crucial, such as healthcare or finance.

Author Response

  1. Can the authors comment on the issue of Limited and heterogeneous data? Collecting CRM data can be challenging due to the need for invasive or specialized measurements. Additionally, the data may vary across individuals, making it difficult to create a comprehensive and representative dataset.

We thank the reviewer for this comment. It is important to appreciate that the CRM-DL algorithm has been trained to identify varying compensatory status across individuals because it is based on learning from a comprehensive and representative database of more than 650,000 arterial pressure analog waveforms captured in real time from noninvasive photoplethysmogram signals. As a result, the algorithm is capable of assessing the complexity of the compensatory reserve for predicting the onset of hemodynamic decompensation with highly consistent accuracy, sensitivity and specificity as reflected by a Receiver Operating Characteristic (ROC) Area Under the Curve (AUC) of greater than 0.90 [ref. 22].

  1. How does the authors assess a reasonable interpretability? Machine learning models used in CRM may produce accurate predictions, but their black-box nature can make it challenging to interpret the underlying factors contributing to those predictions. This can be a concern, especially in healthcare settings where interpretability and explainability are crucial.

Aspects of machine learning models will always have some level of black-box nature, but there is a difference between a deep learning model interpreting an entire waveform through unknown mechanisms vs. a machine learning model only using 10 extracted features. This methodology used in this manuscript improves on explainability as the exact waveform features being tracked can be communicated. However, explainability can be further improved by knowing how trends in these features correspond to CRM and connecting them more effectively to the underlying physiology.

  1. Can the authors comment on new feature selection or the given selection is absolute? Determining the most relevant features or physiological parameters to include in the model can be complex. It requires a deep understanding of the underlying physiology and may involve domain experts to guide the feature selection process. This is important when looking at the differences.

The selection of the most relevant features is based on the fundamental physiological understanding that the integration of all compensatory mechanisms is represented by the morphology of each arterial waveform that consists of two primary waves: (1) an ‘ejected’ wave with features that are dictated by all compensatory mechanisms that influence myocardial function; and (2) a ‘reflective’ wave with features that are influenced by all compensatory mechanisms involved in the control of peripheral blood flow [see refs. 6, 7, 22]. For the CRM-ML model, we realize that a stronger physiological meaning is needed for the selected ten features and highlight this limitation in the discussion on line 449.

  1. 171, vs 0.159, close enough, but still room for improvement.

We agree with the reviewer and proposed some approaches that can be taken to further improve on the ML model on line 452.

  1. Can the authors give comments on generalization? Ensuring that a machine learning model trained on a specific CRM dataset can generalize well to different populations, demographics, or clinical scenarios is crucial. The model needs to account for the variability across individuals and adapt to new data distributions.

‘Generalization’ is based on the premise that the algorithm can accurately identify the status of the compensatory reserve of individuals who represent a population with a wide range of demographics (i.e., age, height, weight), differences in sex, and tolerances to reductions in central blood volume. For our selected 60 subject subset required for equal subject for each LBNP step, 62% were male and 38% were female, with ages from 18-54 years, height from 152 to 193cm and  weight from 44 to 117 kg. These demographics have been added to lines 181 to 183. In addition, the CRM-DL algorithm has been validated in several clinical scenarios that included trauma patients with hemorrhage, sepsis, internal bleeding, and during orthotopic liver transplant.

  1. In terms of the methodology, bagged decision trees have several advantages, such as reducing overfitting and improving prediction accuracy, however there are also some disadvantages to consider, which would be interesting to hear the thoughts of the authors for the present investigation and scope. How the Increased complexity can be addressed? Bagging can increase the complexity of the model, as it involves training multiple decision trees and combining their predictions. This can make the model more difficult to interpret and understand, which can be a disadvantage in some applications. Can the authors comment on the computationally intensive nature of the method particularly for a militar setting? Bagging can be computationally intensive, especially when dealing with large datasets or a large number of decision trees. This can make the training and prediction time longer, which can be a disadvantage in real-time applications. What are the solutions for bias reduction limitations? While bagging can reduce variance and improve prediction accuracy, it has limited impact on reducing bias. This means that if the underlying decision tree model is biased, the bagged model will also be biased. For the blood pressure data, how can we assess the balance of the data? Bagging can be less effective for imbalanced datasets, where one class is underrepresented. This is because the decision trees may focus more on the majority class, leading to poor predictions for the minority class. Can the authors give some guidelines on Limited interpretability? The combination of multiple decision trees in a bagged model can make it difficult to interpret the importance of individual features and their contribution to the final prediction. This can be a disadvantage in applications where interpretability is crucial, such as healthcare or finance.

We appreciate the feedback on potential advantages and disadvantages for the bagged decision tree ML model and we agree with many of the reviewer comments, but it is important to frame these advantages in the context of comparison to a deep learning AI model which has similar weaknesses at higher severities. We have added an additional discussion paragraph detailing the advantages and disadvantages of the bagged decision tree method on line 403.

  1. 88 missing bracket in 19

Thanks for pointing this out. We have fixed in the new manuscript version